# Reversal of Photoinduced Bending of Crystals Due to Internal Refraction of Light

Stanislav Chizhik * 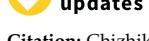, Pavel Gribov, Viktor Kovalskii and Anatoly Sidelnikov

Institute of Solid State Chemistry and Mechanochemistry, SB RAS, 630090 Novosibirsk, Russia
* Correspondence: stas@solid.nsc.ru

**Featured Application: The effect of reversal of photoinduced bending of crystals can be used for light-driven actuators capable of varying the direction of bending controlled by the wavelength.**

**Abstract:** The well-known effect of bending of acicular crystals undergoing photochemical reactions is associated with the transformation gradient across the crystal thickness caused by the absorption of light. It is believed that the direction of bending is unambiguously dictated by the sign of the axial component of the reaction strain and due to the higher light intensity on the irradiated side of the crystal. In this study, it is shown experimentally and theoretically that thin crystals with a convex irradiated surface bend in the opposite direction if their thickness is less than the light penetration depth. The reversal of the bending direction is due to the lens effect, able to overcome the absorption of light in a thin crystal and provide a higher light intensity on the rear side of the crystal. A crystal of $[Co(NO_3)_5NO_2]Cl(NO_3)$ experiencing nitro-nitrito photoisomerization, begins to bend in the opposite direction after it is thinned by etching to 2 μm when irradiated at the wavelengths of 523 nm, 465 nm, and 403 nm, absorbed at a depth of more than 10 μm in the crystal, but bends in the normal direction under 350 nm light absorbed at a depth of about 1 μm. The experimental results are fully confirmed by modeling the interaction of plane EM wave with crystals of various cross sections.

**Keywords:** photomechanical effect; photoinduced bending of crystals; wavelength dependence of photomechanical effect; absorption of light; refraction of light

## 1. Introduction

The phenomena of the mechanical responses of crystals to various external stimuli such as light irradiation, heating, changes in humidity, etc., have become the subject of intensive studies in the last decade [1–13] due to the possibility of using them as microscopic actuators converting signals from external influences into mechanical work [10,12,14,15]. All such phenomena are based on inhomogeneous strains occurring in crystals as the result of inhomogeneous transformation. A gradual chemical transformation caused by the external factors leads to a smooth change in the shape of crystals, such as bending or twisting, going in parallel with the transformation, while phase transitions can lead to spontaneous relaxation acts, accompanied by sharp movements or the destruction of crystals [1,5,7,16–22].

The bending of acicular crystals undergoing photochemical transformations, such as photoisomerization of molecules, observed when crystals are irradiated from one side with the light of a suitable wavelength, is one of the most thoroughly studied phenomena of mechanical response, which is due to the high predictability and reproducibility of the photoinduced bending [1,7,10,12,16,18,23–25]. If the isomerization is reversible with a change in temperature or wavelength, and if the accompanying change in the lattice parameters does not exceed a threshold (a few percent, normally) sufficiently thin crystals can bend and unbend reproducibly many times without noticeable accumulation of residual plastic deformation, destruction or amorphization [10,12,16,20,26]. Such reproducibility



provides an excellent opportunity for their study and quantitative description of the transformation and mechanical response [24], and is also attractive for possible practical applications of the effect.

The macroscopic deformation of bending crystals, including longitudinal elongation/shortening and bending of the crystal axis, is completely determined by the distribution of the axial component of lattice deformation in the crystal cross section [1,24]. The curvature of a bent crystal (the value inverse to the radius of curvature $R$) is determined by the moment of distribution of axial strain $\varepsilon(\mathbf{r})$ over the cross section of the crystal $S$, according to Equation (1);

$$\frac{1}{R} = -\frac{1}{J}\int_S x\varepsilon(\mathbf{r})ds, \tag{1}$$

while the elongation is defined by the average axial strain in the cross section, Equation (2);

$$\frac{\Delta L}{L} = \frac{1}{S}\int_S \varepsilon(\mathbf{r})ds \tag{2}$$

Equations (1) and (2) are a generalization of the expressions obtained in [1] for the case of an arbitrary shape of the cross section of crystals. The value of the static moment of inertia of the crystal section $J$ is determined by Equation (3);

$$J = \int_S x^2 ds \tag{3}$$

where $x$ is the distance from the neutral plane of the bent crystal.

The sign in Equation (1) is taken in such a way that the curvature $1/R$ is positive with a positive transformation strain, $\varepsilon > 0$, which decreases deeper from the irradiated surface, $d\varepsilon/dx < 0$. The latter condition is a natural consequence of light absorption in the bulk of the crystal, leading to a faster transformation near the irradiated crystal surface and, as a consequence, to the appearance of a negative gradient of the transformation degree in the direction of light propagation, $dC/dx < 0$. This condition is considered to be always satisfied, which leads to an unambiguous direction of crystal bending: in the direction of light propagation if $\varepsilon > 0$, or towards the light source if $\varepsilon < 0$.

It should be noted that the strains caused by irradiation can be associated not only with a photochemical transformation, but also, at least, with the direct heating of the crystal, since some part of the energy of the absorbed light is converted into heat. The corresponding contribution to $\varepsilon$ is determined by the value and sign of the coefficient of thermal expansion (CTE), which is most often positive. This phenomenon was described in [27,28], called there the photothermal effect. Since the CTE value usually does not exceed $10^{-4}$ K$^{-1}$, corresponding strains caused by usual light sources turn out to be one or two orders of magnitude lower than the strains caused by photochemical transformations. For this reason, the photothermal bending is usually masked by the stronger effect from photoisomerization. Due to the fact that photothermal strains and their change with time are associated not with the degree of transformation, but with the resulting temperature profiles in the bulk of the crystal, the photothermal bend is clearly observed only on crystals whose thickness significantly exceeds the depth of light penetration. In this case, the weak contribution to the crystal bending from the transformation of a thin surface layer, the thickness of which slowly increases according to the transformation kinetics, can be surpassed by a stronger bending from the temperature gradient, which appears and develops much faster, according to the kinetics of heat transfer from the surface heated by the irradiation. In this case, the direction of bending itself can differ in sign if CTE and transformation strains have opposite signs. This was confirmed in [29], where rapid but weak reversal bending at angles less than 1° was observed on thick salicylideneaniline crystals. In the case of thin crystals considered in this study, the heat releases more uniformly in the bulk of the crystal, which further reduces temperature gradients and the contribution of the photothermal effect, which, therefore, can be neglected here.

For a more explicit formulation of the direction of crystal bending caused by photochemical reaction, one can consider an expression for the crystal bending rate $dR^{-1}/dt$. For the cases of small transformation strain we can use the linear approximation of the interconnection between the strain and the transformation degree [24];

$$\varepsilon(\mathbf{r}) \ = \ \varepsilon_0 C(\mathbf{r}), \tag{4}$$

where $\varepsilon_0$ is a linear coefficient of the transformation strain. Then the bending rate is defined by Equation (5);

$$\frac{dR^{-1}}{dt} \ = \ -\frac{\varepsilon_0}{J} \int_S x \frac{dC(\mathbf{r})}{dt} ds. \tag{5}$$

The transformation kinetics can be described as a first order reaction, according to Equation (6) [1,24];

$$\frac{dC(\mathbf{r})}{dt} \ = \ I_0 \varphi \mu v \frac{I(\mathbf{r})}{I_0}(1 - C(\mathbf{r})) \tag{6}$$

$I_0$ is the intensity of the light incident on the crystal in terms of photons flux (photon*cm$^{-2}$*s$^{-1}$), $I(\mathbf{r})/I_0$ is the bulk distribution of the relative intensity of light inside the crystal, $\varphi$ is the quantum yield of the photochemical reaction, $\mu$ is the absorption coefficient in the Bouguer-Beer-Lambert (BBL) law (cm$^{-1}$), $v = 1/C_0$ the volume per one molecule in the crystal (cm$^3$), $C_0$ is molecule concentration (cm$^{-3}$). Then the starting bending rate $\Omega_0$ is defined by Equation (7);

$$\Omega_0 \ = \ \frac{dR^{-1}}{dt}\bigg|_{t=0} \ = \ -\frac{\varepsilon_0 I_0 \varphi \mu v}{J} \int_S x \frac{I(\mathbf{r})}{I_0} ds, \tag{7}$$

from which it follows that the direction of the bend, that is, the sign of $\Omega_0$, is uniquely determined by the sign of the deformation coefficient $\varepsilon_0$ and the sign of the relative intensity moment, the integral in Equation (7). The latter is normally considered negative as the absorption of light should lead to a negative intensity gradient $dI/dx < 0$.

Equation (7) has a fairly general meaning and allows us to analyze situations in which the direction of crystal bending can change to the opposite if the integral in Equation (7), the moment of distribution of relative light intensity in the crystal cross section, changes its sign for some reasons that turn out to be stronger than the absorption law. For example, this makes it possible to analyze the cases of mechanical response of extremely thin crystals, the thickness of which is much less than the characteristic light attenuation depth $\mu^{-1}$, or even less than the wavelength. Such cases arise in experiments with crystals of submicron thickness, but until now it was believed that the direction of crystal bending is dictated only by the light absorption and the sign of $\varepsilon_0$. Other factors may be related to various optical effects such as refraction, diffraction and interference of light in crystals.

The purpose of this study is to test the influence of crystal thickness on the observed direction of photoinduced bend. To solve this problem, an experimental study of $[Co(NH_3)_5NO_2]Cl(NO_3)$ thin crystal bending under various light sources and a theoretical consideration of the interaction of an electromagnetic wave with a thin crystal absorbing the light were carried out. The choice of $[Co(NH_3)_5NO_2]Cl(NO_3)$ is due to the very good knowledge of the linkage isomerization reaction occurring in it, which is accompanied by a reproducible bending of needle-shaped crystals. This coordination compound exists in two isomeric forms, differing in the coordination of the metal by the $NO^{2-}$ ligand: Co-NO$_2$ (nitro) and Co-ONO (nitrito). The transition from the thermally stable nitro form to the nitrito form occurs as a photochemical isomerization upon exposure of crystals to UV or visible light at $\lambda < 530$ nm [30–33]. The nitrito form is kinetically stable at temperatures below room temperature; the reverse transformation to the nitro form takes about several months at 0 °C. Heating the nitrito form to 80 °C reduces the recovery time to a few minutes [24,31,32,34–38]. Bending and elongation of crystals occurs as a result of an increase in

the lattice parameter *b* (~4% increase on nitro-nitrito transition at 0 °C) oriented along the long axis of the crystal.

## 2. Materials and Methods

To study the photoinduced bending of the crystal, a thermostated chamber with optical windows was used, mounted on Neophot 21 (Carl Zeiss) inverted optical microscope. The crystal, cantilevered to a thin glass holder, was placed in the chamber in a flow of dry nitrogen at a temperature of 0 °C maintained with an accuracy of ±1 °C (detailed scheme is described in [39]). To carry out photoisomerization, the crystal was preheated at 80 °C and cooled down to 0 °C, and then irradiated with LED sources with $\lambda$ = 523 nm, 465 nm, 403 nm and a high-pressure xenon arc lamp with a 350 nm bandpass filter.

During the transformation, time lapse images of the crystal were recorded with a digital camera, using additional 625 nm LED source for crystal illumination, which does not cause photochemical transformations. To determine the time dependence of the crystal curvature the time lapse series were analyzed using specially developed software, "Bending crystal track" (http://imagej.net/PhotoBend (accessed on 21 November 2022)), a plugin for ImageJ, image analysis program [40]. The method is based on the recognition of crystal image elements and determination of the coordinates of three specific points on its axis: two located at the ends of the crystal and one in the middle, from which the current length and curvature of the crystal are calculated [24].

In order to obtain a thin crystal suitable for studying in the setup described above, the method of dissolving a thicker crystal attached to a holder was used. The initial crystal of the complex, having an approximately square cross section ~10 μm thick, was glued to a glass capillary with a thin end (~40 μm in diameter). A cuvette with $HCONH_2$ (formamide) used to thin the crystal was placed under optical microscope. Since $[Co(NO_3)_5NO_2]Cl(NO_3)$ dissolves much more slowly in formamide than in water, formamide can work as a polishing etchant for these crystals. Another cuvette with dried dimethyl sulfoxide (DMSO) was placed nearby, which was used to quickly interrupt dissolution and wash the crystal from formamide. The crystal on the holder was carefully immersed in formamide, while rotational, vertical and horizontal movements were performed with constant monitoring of the dissolution process under the microscope. Dissolution occurs most rapidly from the edges of the crystal, and it acquires a rounded shape in the cross section. Periodically, the dissolution process was stopped by immersion in DMSO to inspect and control the result. If necessary, the procedure is repeated until the desired thickness and geometry quality is achieved. Using this technique, it was possible to obtain a crystal sample with a thickness of ~2 μm, shown in Figure 1. The natural loss of the faceting of crystals during dissolution results in rounded shape of the crystal's cross-section, which became a critical feature of this method for the experimental result obtained in the study. An example of a cross section image of a crystal that acquires a rounded shape after the etching is shown in Figure S1 of the Supplementary Materials.

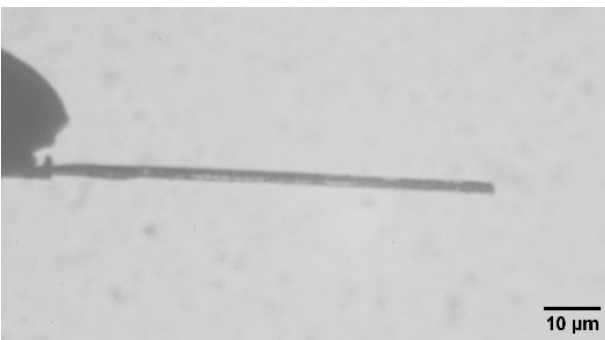

**Figure 1.** Optical microscopic image of the 2 μm thick $[Co(NH_3)_5NO_2]Cl(NO_3)$ crystal obtained by etching of a thicker crystal cantilevered to a holder (the edge of the glass capillary can be seen on the left side of the image).

### 3. Experimental Results

In all previous studies of photoinduced bending of $[Co(NH_3)_5NO_2]Cl(NO_3)$, crystals with a thickness of more than ~20 μm were used, and bending away from the light source was always observed, which corresponds to the positive curvature, according to the accepted convention about the curvature sign. This fully agrees with Equation (7), taking into account the negative light intensity gradient $dI/dx < 0$ due to its absorption in the bulk of the crystal [24,35,36,39]. The light penetration depth in this substance, ~10 μm at $\lambda = 465$ nm, and ~30 μm at $\lambda = 403$ nm and at 523 nm, always turned out to be less or slightly more than the thickness of the crystals used. As a consequence, the light absorption was the main factor determining the distribution of light intensity in the volume of the crystal. Therefore, in order to experimentally detect the effect of any other factors, it was necessary to use crystals with a thickness much smaller than the characteristic absorption depth, $h\mu << 1$. The thickness of the crystal obtained by dissolution is at least five-times less than the characteristic absorption depth for the most strongly absorbed light in the visible range, 456 nm.

For all three sources of visible light, it turns out that the studied crystal bends in the opposite direction, compared to ordinary thicker crystals. At the same time, the crystal lengthens by $\varepsilon_0 \sim 4\%$ in all cases, which confirms that we are dealing with the same substance experiencing nitro-nitrito isomerization. An example of a combined image of a crystal in three successive moments of the process at $\lambda = 465$ nm is shown in Figure 2. Since crystal bending is strongly associated with an inhomogeneous distribution of the degree of transformation, then from Figure 2 follows the conclusion about the abnormal situation when the opposite side of the crystal isomerizes faster than the irradiated surface. The only possibility of such an effect must be associated with some factors that lead to the concentration of light intensity on the reverse side of the crystal. The most natural assumption is the lens effect, the refraction of light inside the crystal, which turns out to be stronger than absorption for a sufficiently thin crystal.

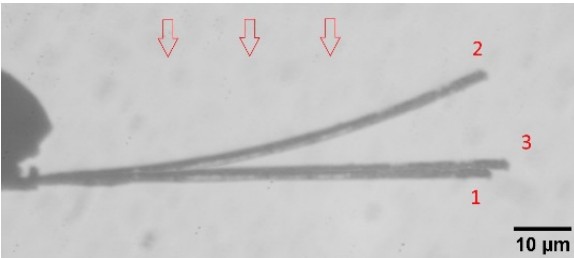

**Figure 2.** Combined image of the crystal at three different moments of transformation at $\lambda = 465$ nm: (1)—initial state of the crystal, (2)—position of maximum bend (irradiation for 3 min), (3)—final shape of the crystal (after 36 min of irradiation). The arrows indicate the direction of light.

To test this hypothesis, an experiment was carried out with the UV light source $\lambda = 350$ nm. According to the absorption spectra of $[Co(NH_3)_5NO_2]Cl_2$ solution given in [31], the extinction of this compound at 350 nm is approximately an order of magnitude higher than at 465 nm. Therefore, the characteristic absorption depth can be assessed to ~1 μm at 350 nm. Three successive states of the crystal under the UV light source are combined in Figure 3: the initial, the maximum bend at the intermediate transformation, and the completely transformed crystal (the latter is shifted vertically not to overlap the original crystal). As can be seen, the direction of crystal bending returns to the normal when using strongly absorbed 350 nm light. It is also seen that the crystal bend amplitude turned out to be much stronger with the UV, compared to $\lambda = 465$ nm. This means that the absorption of light has a greater effect on the intensity non-uniformity than the expected refraction. The free end of the crystal left the focal plane when bent, so that the absolute values of the curvature are unreliable. Nevertheless, the theoretical estimate of the

minimum radius of curvature for this crystal at $\varepsilon_0 = 0.04$ and $\mu^{-1} = 1$ μm is about 70 μm, according to [1], which is consistent with the observation.

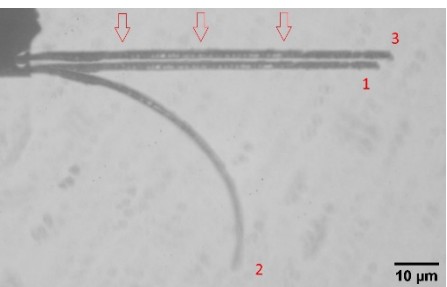

**Figure 3.** Combined image of the crystal at three different moments of transformation at $\lambda = 350$ nm: (1)—initial state of the crystal, (2)—position of maximum bending (after 15 min of irradiation), (3)—final shape of the completely transformed crystal (80 min of irradiation, the image is shifted vertically up). The arrows indicate the direction of light.

The obtained result of changing the direction of the crystal bend with a change in the irradiation wavelength is phenomenologically similar to the phenomenon observed in diarylethene crystals [11], when a change in the wavelength from 365 nm to 380 nm also led to a change in the bend direction to the opposite, but was observed only at some initial stage of transformation. It is noteworthy that, similar to the effect discussed here, the 380 nm radiation is also characterized by weaker absorption in diarylethene crystals compared to the 365 nm radiation [11]. As the explanation of the photoinduced bending is based on the traditional consideration of BBL law in [11], the authors assume a different sign of strain $\varepsilon_0$ at a small degree of transformation as the reason of the bend direction reversal; when the strain sign becomes usual upon further transformation, the direction of the bend returns to normal. The results of the study proposed here offer a different look at the effect observed in [11].

The dependences of the crystal curvature on time are shown in Figure 4. Information of primary interest is in the sign of the curvature, as well as in the magnitude of its change, while the characteristic duration of the process is determined mainly by the intensity of a particular source and the quantum yield of the reaction at the corresponding wavelength. The largest bend reversal effect was obtained for 403 nm light source, since light absorption by the substance is the weakest at this wavelength, and the expected refractive factor should be most pronounced. The lower magnitude of the effect at 465 nm is explained by the higher absorption at this wavelength. The absorption of 523 nm light by the initial nitro isomer is comparable to the absorption at 403 nm, thus a similar bending magnitude would be expected for these light sources. However, the transformation to the nitrito form leads, according to [31,39], to an approximately twofold increase in the extinction at 523 nm, so the effect of refraction is reduced for this light source. The low conversion rate at $\lambda = 523$ nm is due to low quantum yield, approximately an order of magnitude lower than at 403 nm [39].

The assumption about the influence of refraction providing the concentration of light on the back side of the crystal due to the lens effect requires theoretical verification. It is intuitively clear that the lens effect should require convex shape of the irradiated crystal surface, while light falling on a flat crystal face should not lead to its concentration in the bulk. However, the diffraction effects may play important role for crystals of wavelength range thickness, independently of the cross-section shape. For this, a direct calculation of the electromagnetic wave penetrating crystals of various cross sections was carried out for various ratios of crystal thickness, wavelength and absorption depth.

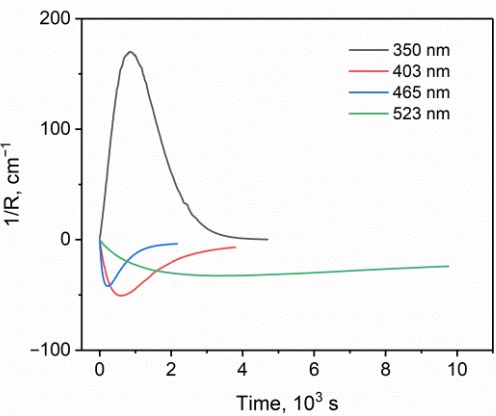

**Figure 4.** Dependences of the crystal curvature on the transformation time for different light sources.

## 4. Model of the Interaction of Light with a Crystal

### 4.1. Simple Qualitative Model

The influence of light refraction on the direction of the crystal bend can be qualitatively explained using the geometrical optics approximation shown in Figure 5.

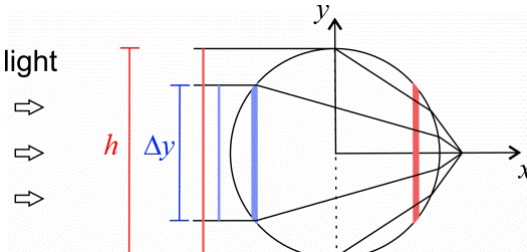

**Figure 5.** Qualitative scheme of light propagation in a crystal of round cross section.

If light falls on a crystal having a convex surface, then the layers located on opposite sides of the neutral plane (vertical dashed line in Figure 5) receive different amounts of light flux. Layers located closer to the irradiated surface (layer highlighted in blue) receive a portion of the light flux proportional to their height $\Delta y$. Layers behind the neutral line, such as the red layer in Figure 5, receive almost the entire luminous flux incident on the crystal. If we neglect the effects of reflection and absorption, then the average light intensity in the layers in front of the neutral plane is approximately equal to $I_0$, while in the layers behind the neutral plane it is equal to $h I_0 / \Delta y$. That is, the light intensity is approximately constant in the front half of the crystal section, and increases with $x$ in the back half, resulting in the negative bend direction, according to Equation (7). A slight absorption of light will lead to definite decrease of the light intensity with $x$, which will lead to a decrease in the magnitude of intensity moment in Equation (7). At some critical value of the absorption coefficient, the intensity moment becomes zero and the crystal will not bend during the transformation. At a higher absorption coefficient, the crystal will bend in the usual direction, given by the sign of $\varepsilon_0$ and the light absorption law. To accurately describe this phenomenon, taking into account all the effects of wave optics, including reflection, refraction, diffraction, and interference, it is necessary to directly simulate the interaction of a plane wave with a crystal.

### 4.2. Theoretical Formulation and Calculation Method

To simulate wave propagation in a circular crystal, one can use the solution of the problem for the case of normal incidence of light on an infinite round cylinder, similar to the well-known Mie theory for light scattering by spherical particles [41–43]. The general solution is represented as a superposition of solutions corresponding to two different

polarizations of the incident wave. In the case of $s$-wave (the electric field of the original wave is perpendicular to the plane of incidence) the electric field vector **E** is parallel to the crystal axis in the whole space. In the case of $p$-wave (the electric field vector in the plane of incidence), the electric field of the initial wave is polarized perpendicular to the crystal axis, while the magnetic field vector **H** is parallel to the crystal axis in the whole space. Further calculations using this model show that the intensity distributions in the crystal cross section obtained for different polarizations lead to qualitatively similar results for the relative intensity moment in Equation (7) in all cases considered. The insignificant quantitative difference is associated with the difference in the reflection coefficients for $s$ and $p$ waves at oblique incidence of the wave on the interface between the media [42]. Therefore, to shorten the discussed results, we will analyze only the $s$-wave case, which turns out to be less demanding on computational resources when solving the problem numerically.

The complex amplitude of $z$ component of the electric field for the $s$-wave case is determined by the following expressions in a cylindrical coordinate system $(r,\theta)$. The electric field inside the crystal $E_{\mathrm{int}}$ is defined by

$$
\begin{aligned}
E_{\mathrm{int}}(r,\theta) &= E_0 \sum_{n=-\infty}^{\infty} F_n(\theta) d_n J_n(mk_0 r) \\
F_n(\theta) &= (-i)^n e^{in\theta} \\
d_n &= \frac{J_n{}'(k_0 a) H_n(k_0 a) - J_n(k_0 a) H_n{}'(k_0 a)}{m J_n{}'(mk_0 a) H_n(k_0 a) - J_n(mk_0 a) H_n{}'(k_0 a)}
\end{aligned}
\tag{8}
$$

where $E_0$ is the amplitude of the plane wave normally incident on the crystal in Figure 5;

$$
E_{\mathrm{inc}} = E_0 e^{ik_0 x},
\tag{9}
$$

characterized by the wave vector $k_0 = 2\pi/\lambda$ and wavelength $\lambda$ in vacuum, $m$ is the complex refractive index of the crystal (the crystal is considered non-magnetic);

$$
m = \sqrt{\epsilon' + i\epsilon''},
\tag{10}
$$

determined by the real and imaginary components, $\epsilon'$ и $\epsilon''$, of the relative complex permittivity $\epsilon = \epsilon' + i\epsilon''$, $J_n$ and $H_n$ are the Bessel and Hankel functions of the first kind of order $n$, primes denote the derivative.

The field of the scattered wave $E_s$ is determined by Equation (11);

$$
\begin{aligned}
E_s(r,\theta) &= -E_0 \sum_{n=-\infty}^{\infty} F_n(\theta) b_n H_n(k_0 r) \\
b_n &= \frac{m J_n{}'(mk_0 a) J_n(k_0 a) - J_n(mk_0 a) J_n{}'(k_0 a)}{m J_n{}'(mk_0 a) H_n(k_0 a) - J_n(mk_0 a) H_n{}'(k_0 a)}
\end{aligned}
\tag{11}
$$

To calculate the field, we used the constant value of the real component of relative permittivity $\epsilon' = \epsilon_0 = 2.5$, which corresponds to the refractive index $n_0 = \epsilon_0{}^{1/2} \sim 1.58$, typical for inorganic glasses. To simulate a substance with different absorption, the imaginary component of relative permittivity $\epsilon''$ was varied to obtain the required values of the absorption coefficient $\mu$, determined by Equation (12);

$$
\mu = 2k_0 \mathrm{Im}(m).
\tag{12}
$$

The intensity of the wave (energy flux) is determined by the time-averaged module of the Poynting vector, and thus can be defined using the calculated electric field according to Equation (13) [42];

$$
I = \frac{1}{2\eta_0} \mathrm{Re}(m)|\mathbf{E}|^2,
\tag{13}
$$

where $\eta_0$ is the impedance of free space. Thus, the relative intensity appearing in Equation (7) is defined as;

$$\frac{I}{I_0} = \text{Re}(m)\left|\frac{\mathbf{E}}{E_0}\right|^2. \tag{14}$$

In the case of non-round cross section one can find a numerical solution of the wave equation. Due to the fact that $\epsilon$ does not depend on $z$, the formulation of the problem for $z$ component of the field in the $s$-wave case consists in solving the wave Equation (15);

$$\Delta E + k_0^2 \epsilon E = 0, \tag{15}$$

with $\epsilon = 1$ outside the crystal. A continuous solution of Equation (15), which satisfies the stated conditions, automatically satisfies the boundary condition for the continuity of the tangential component of the electric field at the crystal boundary.

The numerical solution was obtained using the finite element method implemented in FreeFEM++ [44]. To implement the calculations, the total field was represented as a superposition of the external wave field and the perturbation field $E = E_{\text{inc}} + E_{\text{ex}}$. Since $E_{\text{inc}}$ identically satisfies the wave Equation (15) outside the crystal, Equation (15) takes the form of Equation (16);

$$\Delta E_{\text{ex}} + k_0^2 \epsilon E_{\text{ex}} = \rho(r)k_0^2(1-\epsilon)E_{\text{inc}} \tag{16}$$

which allows one to calculate the unknown perturbation field $E_{\text{ex}}$ for a given field of the initial wave $E_{\text{inc}}$. The function $\rho(r)$ determines the region of space occupied by the crystal;

$$\begin{aligned} \rho(r) &= 1 \text{ inside crystal} \\ \rho(r) &= 0 \text{ outside crystal} \end{aligned} \tag{17}$$

The calculations were carried out in a cylindrical region 40 μm in diameter surrounding the crystal. To avoid parasitic reflections of the scattered wave from the artificial boundaries of the finite calculation region and to simulate the unimpeded propagation of the scattered wave into infinite space, a 5 μm cylindrical layer was defined on the outer boundary of the computational region with the imaginary component of the permittivity gradually increasing with distance from the crystal, which ensures reflectionless attenuation of the scattered wave.

### 4.3. Simulation Results

4.3.1. Round Cross Section

Figure 6 visualizes the relative radiation intensity $I/I_0$, calculated for the wave defined in Equations (8) and (11) for two cases that reproduce the experimental conditions of irradiation of the 2 μm thick crystal by the light sources $\lambda$ = 400 nm (absorption depth $\mu^{-1}$ = 30 μm) and $\lambda$ = 350 nm ($\mu^{-1}$ = 1 μm). The internal oscillating structure of the intensity is associated with the interference of internal reflected waves: the maxima and minima of the intensity alternate with the period of half the wavelength in the substance, $\lambda/(2\epsilon_0{}^{1/2})$. In addition, the resulting interference pattern gives an idea of the shape of the wave fronts inside the crystal, visualizing the condensation of light as a result of refraction. Strictly speaking, such an interference pattern is valid for a monochromatic source. However, with a typical spectrum width of the used sources $\Delta\lambda \sim 0.03\lambda$ [39] the characteristic coherence length is $\sim 30\lambda$. That is, the calculation result for a monochromatic source quite correctly shows the real intensity distribution for crystals thinner than 10 μm in the case of using light sources like LED for irradiation. This means that in addition to the effects discussed in this study, sufficiently thin crystals should also contain a fine structure of the transformation rate.

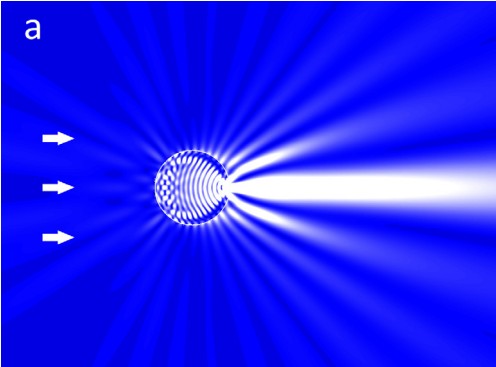
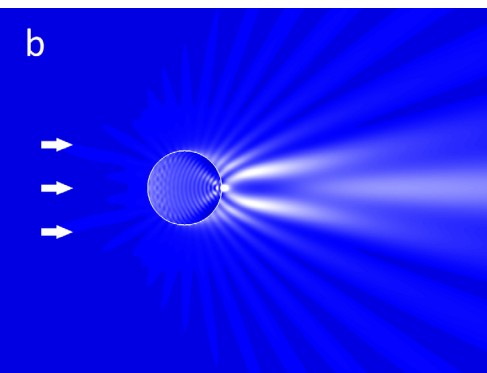

**Figure 6.** Visualization of the intensity distribution of a wave scattered by a 2 μm thick crystal for two cases considered in the experiment: (**a**) $\lambda$ = 400 nm, $\mu^{-1}$ = 30 μm, (**b**) $\lambda$ = 350 nm, $\mu^{-1}$ = 1 μm. The arrows show the direction of the incident light.

Figure 7 shows 3D surfaces of $I/I_0$ inside the crystal for these two cases. As can be seen, in the case of weak absorption ($\lambda$ = 400 nm, Figure 7a) there is a tendency for a significant increase in the intensity towards the rear surface of the crystal, associated with the condensation of the light flux. For the case of strong absorption ($\lambda$ = 350 nm, Figure 7b), there is a regular decrease in the intensity deep into the crystal. For a clearer idea of how the obtained distributions form the intensity moments of different signs, one can consider the dependences of the radiation power incident on the flat layers of matter inside the crystal (layers shown in blue and red in Figure 5), depending on the distance to the neutral surface $x$. The value of the relative power, normalized to the power incident on the entire crystal, is determined by the intensity integral over the layer height, Equation (18);

$$\frac{P(x)}{P_0} = \frac{1}{h}\int_{\Delta y} \frac{I(x,y)}{I_0}\,dy. \tag{18}$$

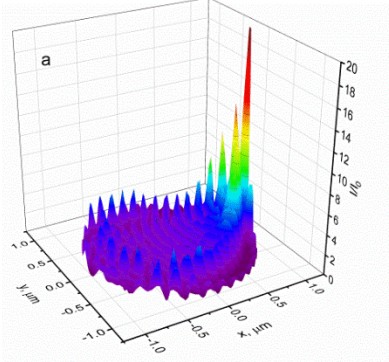
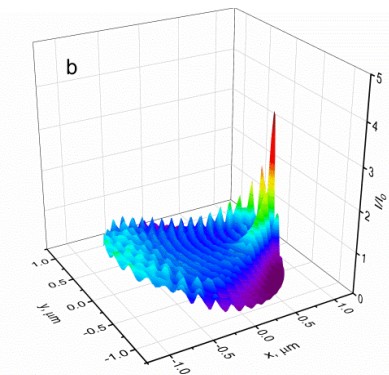

**Figure 7.** 3D light intensity distributions inside the crystal for the two cases shown in Figure 6: (**a**) $\lambda$ = 400 nm, $\mu^{-1}$ = 30 μm, (**b**) $\lambda$ = 350 nm, $\mu^{-1}$ = 1 μm.

Relative power graphs are shown in Figure 8. In the case of weak absorption, the power increases with $x$ in the front half of the crystal and saturates in the back half, as was qualitatively predicted above, which leads to a faster conversion on the back side of the crystal. In the case of strong absorption, the radiation power maximum is localized near the front surface of the crystal, as is usually assumed in all works studying the photoinduced bending of crystals.

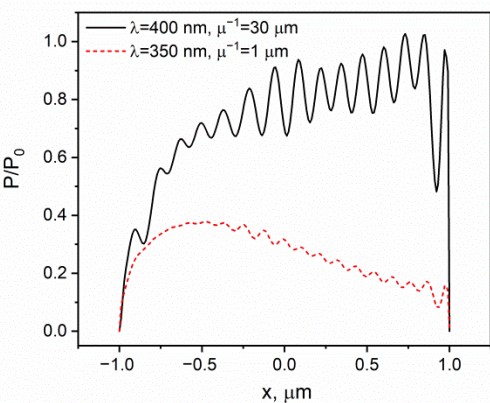

**Figure 8.** Relative power of radiation incident on vertical sections of a crystal at different depths.

Figure 9 shows the results of calculations of the reduced moment of intensity $\Omega_r$

$$\Omega_r = \frac{\Omega_0}{\varepsilon_0 I_0 \varphi \mu \upsilon} \frac{J}{h^3} = -\frac{1}{h^3} \int\limits_S x \frac{I(x,y)}{I_0} ds, \tag{19}$$

a dimensionless characteristic, convenient for comparing the results depending on the product $\mu h$, which characterizes the absorption of light by a crystal with a diameter $h$. The results are obtained for $\lambda = 400$ and crystals of various thicknesses, from 12 nm to 5 $\mu$m, in order to show the dependence of the effect of bend direction reversal on the diffraction parameter $h/\lambda$. Positive values of $\Omega_r$ correspond to the normal direction of crystal bend, which follows from the BBL law, while negative values correspond to the reversal of the bend direction associated with the refraction of light in the crystal. As can be seen, for all thicknesses of the crystal, a general qualitative dependence is observed—the direction of the bend is reversed when the absorption parameter $\mu h$ decreases below a certain value, which increases with increasing influence of diffraction (with a decrease in $h/\lambda$). In the region of the normal direction of the crystal bend, the dependences $\Omega_r(\mu h)$ behave systematically at different $h/\lambda$: after passing through a maximum at $\mu h > 4$, $\Omega_r$ asymptotically tends to 0 at strong absorption, when the transformation occurs only in a thin near-surface layer with a thickness of $\sim \mu^{-1}$. Increasing the diffraction effects increases the minimum required absorption for the normal bend direction, i.e., diffraction enhances the role of refraction in the area of balance conditions between various factors that determine the resulting bend direction. At the same time, in the range of negative values of $\Omega_r$, quantitative dependence of the effect on diffraction parameter $h/\lambda$ has a complex character. When approaching the diffraction limit $h/\lambda \sim 1$ the slope of $\Omega_r(\mu h)$ oscillates with $h/\lambda$, which means that the effect of reversal of intensity distribution periodically increases and weakens with $h/\lambda$.

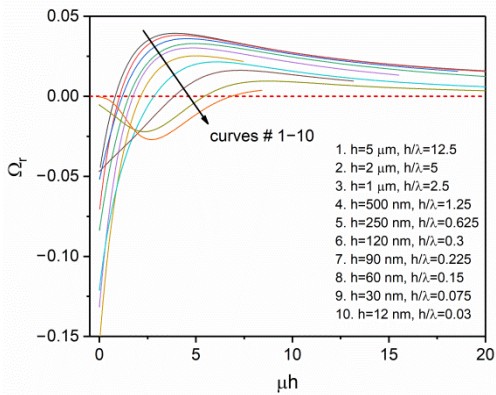

**Figure 9.** Dependences of the reduced moment of intensity $\Omega_r$ (Equation (19)) on the parameter of light absorption in a crystal $\mu h$ for crystals of different thicknesses ($\lambda = 400$ nm).

This effect is associated with the successive localization of the interference maximum or minimum near the rear surface of the crystal. To demonstrate this effect, Figure 10 shows an example of the dependence of $\Omega_r$ on the crystal thickness (in units of the diffraction parameter inside the crystal $\text{Re}(m)h/\lambda$ at a constant absorption coefficient corresponding to an absorption depth of 10 μm and $\lambda$ = 400 nm in vacuum. Noticeable oscillations of $\Omega_r(h)$ begin at $\text{Re}(m)h/\lambda < 10$ ($h < 2.5$ μm in the example under consideration) and are characterized by a period approximately equal to half the wavelength in the crystal $\Delta h = \lambda/(2\text{Re}(m))$. At $h\text{Re}(m)/\lambda < 1/2$ the crystal becomes so thin that the field can be considered almost uniform over the entire thickness of the crystal, which leads to a zero moment of intensity distribution, $\Omega_r = 0$. That is, in the limit of $h/\lambda \ll 1$ and $\mu h \ll 1$ diffraction completely suppresses all effects that cause crystal bending; nanosized crystals should transform without bending, if the substance is not characterized by too much absorption.

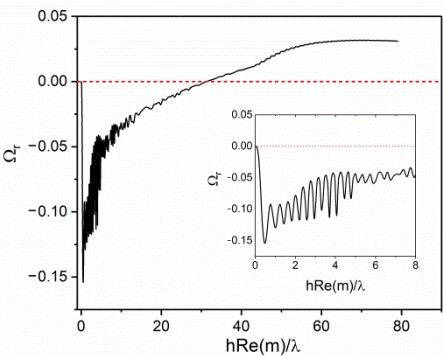

**Figure 10.** Dependence of $\Omega_r$ on crystal thickness at constant absorption $\mu^{-1}$ = 10 μm for $\lambda$ = 400 nm.

### 4.3.2. Square Cross Section

To check the influence of the shape of the crystal's cross section and the orientation of its faces to the light, a numerical solution of the problem was obtained for a square crystal in the case of normal light incidence on a flat face (square orientation), or light incidence on the side edge of the crystal, along the diagonal of the cross section (diamond orientation). Figure 11 shows the dependences $\Omega_r(\mu h)$ for crystals of different thicknesses and orientations towards the light, as well as the theoretical dependence for square orientation, following from BBL law, according to Equation (20);

$$\Omega_{\text{BBL}} = \frac{1}{h^2} \int_0^h \left(\frac{h}{2} - x\right) e^{-\mu x} dx. \tag{20}$$

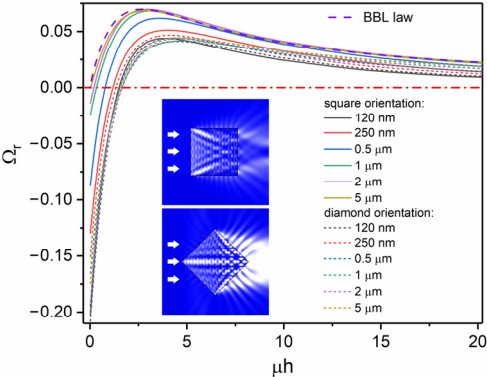

**Figure 11.** $\Omega_r(\mu h)$ dependences for crystals with square-shaped section of different thicknesses and orientation to the light ($\lambda$ = 400 nm). The arrows show the direction of the incident light.

As can be seen, crystals with a thickness of more than 1 μm in a square orientation qualitatively follow the dependence for BBL law, predominantly bending in the usual direction determined by light absorption. Reducing the thickness to values $h \sim \lambda$ leads to the reversal of bend direction, similar to the case of a circular section. This effect is caused by the diffraction of the incident wave on a thin crystal leading to an efficient light penetration from the side faces. The behavior of crystals with a sub-wavelength thickness becomes qualitatively similar to the behavior of crystals in the diamond orientation discussed below. Due to diffraction effects, the wave ceases to distinguish the orientation of very thin crystals.

The diamond orientation is characterized by a strong refraction effect due to the oblique incidence of light on the face of the crystal. As a result, the bend reversal effect is observed, which is even larger in amplitude than in the case of the circular cross section. $\Omega_r(\mu h)$ curves are poorly dependent on crystal thickness for the diamond orientation, which is apparently due to the small contribution of the corner region of cross section to intensity moment, compared to the round shape.

Thus, the same crystals can exhibit a normal or reversed bending direction, depending on the orientation of the crystal to the light, i.e., whether the light is incident on a flat face or on a side edge. Under normal irradiation of wide crystal faces, the usual direction of bending is observed, well described by the standard BBL law, Equation (20).

## 5. Conclusions

The results obtained experimentally and theoretically prove that in the case of thin crystals it is possible to observe the effect of a reverse bend direction in crystals undergoing photochemical transformations. To observe the effect, it is necessary that two conditions are satisfied: (1) the crystals have a convex cross-sectional shape close to round (for example, a hexagon) or are oriented to the light flux with an edge rather than a flat face, (2) their thickness is of the order of the light absorption depth $\mu h \sim 1$ or less. The effect is associated with the refraction of light wave in the cross section of the crystal, leading to a concentration of light intensity near the rear surface of the crystal. Theoretical calculation also shows that when approaching strong diffraction effects at $h/\lambda < 1$, the range of conditions under which bend reversal can be observed expands towards greater absorption (up to $\mu h \sim 10$, in the investigated range of conditions). In this case, an oscillating dependence of the bending rate on crystal thickness should be observed with a period equal to half the wavelength in the substance. In the case of very thin crystals, $h/\lambda \ll 1$, the theory predicts a homogeneous transformation of weakly absorbing crystals ($\mu h \ll 1$).

The results obtained indicate the need to take into account the effects of refraction and diffraction of light in the quantitative description of the process of crystal bending and offer a new explanation for some previously obtained results of reversing the direction of crystal bending when using light that is weakly absorbed by a substance, as in [11]. In addition, the effect predicted can be used in the development of light-active actuators that allow a controlled change in the direction and speed of bending, depending on the wavelength.

**Supplementary Materials:** The following supporting information can be downloaded at: https://www.mdpi.com/article/10.3390/app122312007/s1, Figure S1: cross-section images of $[Co(NO_3)_5NO_2]Cl(NO_3)$ crystal subjected to polishing etching in formamide: faster dissolution of crystal edges, visible in the initial stages of etching (left); rounded cross section achieved with the further dissolution (right).

**Author Contributions:** Conceptualization, S.C. and A.S.; funding acquisition, S.C.; investigation, A.S.; methodology, S.C. and A.S.; project administration, A.S.; resources, P.G. and V.K.; software, S.C.; supervision, S.C.; validation, S.C., P.G. and V.K.; visualization, P.G. and V.K.; writing—original draft, S.C.; writing—review & editing, S.C. and A.S. All authors have read and agreed to the published version of the manuscript.

**Funding:** This research was funded by the Russian Science Foundation, grant number 22-23-01130.

**Conflicts of Interest:** The authors declare no conflict of interest.

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
