# Peer review of "Reversal of Photoinduced Bending of Crystals Due to Internal Refraction of Light"

_applsci, doi:10.3390/app122312007_

Round 1

Reviewer 1 Report

This paper reports on the reversal bending behavior of crystals thin enough by the effect of internal refraction. The internal refraction causes the situation that light intensity at the crystal's rear side is higher than that of the irradiated surface. This effect becomes apparent when using thin crystals and wavelengths of low absorption. They confirmed this by experimentally and theoretically, constructing a new basis for the photomechanical bending of crystals. In my view, the paper can be accepted when just a few minor corrections are made.

1. It seems instruction sentences in the template remain. (lines 119-128)

2. It may be better to mention the photo-thermal effect. I think the effect can be neglected in the case of this crystal, but some recent papers (for example, https://doi.org/10.1039/D2MA00825D) reported that photothermal and isomerization cause different bending directions. The mention of the photothermal effect will clarify the bending of this crystal caused only by photoisomerization.

Author Response

Response to the comments from Reviewer 1

Comment: 1. It seems instruction sentences in the template remain. (lines 119-128)

Response to the comment: We apologize for accidentally leaving part of the text from the template in the manuscript. This part has been removed in the revised version.

Comment: 2. It may be better to mention the photo-thermal effect. I think the effect can be neglected in the case of this crystal, but some recent papers (for example, https://doi.org/10.1039/D2MA00825D) reported that photothermal and isomerization cause different bending directions. The mention of the photothermal effect will clarify the bending of this crystal caused only by photoisomerization.

Response to the comment: We confirm that, just as in the mentioned publication, in experiments carried out about 5 years ago, we observed a similar photothermal effect on thick crystals of the substance under study: crystals 200-300 um thick demonstrate a fast bend at an angle of less than 1 degree away from the source light that occurs in a fraction of a second when the irradiation is turned on. With further irradiation, a slow bending occurred due to photoisomerization (in the same direction, since the transformation in this case causes a positive strain). At the end of irradiation, the crystal abruptly unbends in the opposite direction at the same small angle as at the beginning of the process, and then slowly unbends following the reverse transformation. We also attributed this effect to thermal deformations, but did not publish the result.

Taking into account the low power of the light sources used and the nature of the deformations causing the photothermal effect, it can be observed only on crystals whose thickness is noticeably higher than the depth of light absorption. In our study, we consider the reverse case - crystals with a thickness less than the absorption depth. To explain this point, we have added the following part to the text of the revised manuscript:

“It should be noted that the strains caused by irradiation can be associated not only with a photochemical transformation, but also, at least, with direct heating of the crystal, since some part of the energy of the absorbed light is converted into heat. The corresponding contribution to e is determined by the value and sign of the coefficient of thermal expansion (CTE), most often positive. This phenomenon was described in [27, 28] called there the photothermal effect. Since CTE value usually does not exceed 10-4 K-1, corresponding strains caused by usual light sources turn out to be one or two orders of magnitude lower than the strains caused by photochemical transformations. For this reason, the photothermal bending is usually masked by the stronger effect from photoisomerization. Due to the fact that photothermal strains and their change with time are associated not with the degree of transformation, but with the resulting temperature profiles in the bulk of the crystal, the photothermal bend is clearly observed only on crystals whose thickness significantly exceeds the depth of light penetration. In this case, the weak contribution to the crystal bending from the transformation of a thin surface layer, the thickness of which slowly increases according to the transformation kinetics, can be surpassed by a stronger bending from the temperature gradient, which appears and develops much faster, according to the kinetics of heat transfer from the surface heated by the irradiation. In this case, the direction of bending itself can differ in sign if CTE and transformation strains have opposite signs. This was confirmed in [29], where rapid but weak reversal bending at angles less than 1° was observed on thick salicylideneaniline crystals. In the case of thin crystals considered in this study, the heat releases more uniformly in the bulk of the crystal, which further reduces temperature gradients and the contribution of the photothermal effect, which, therefore, can be neglected here.”

Reviewer 2 Report

This manuscript reported an interesting phenomenon of the mechanical response of crystals toward light irradiation. The corresponding theoretical proposal is of significance in understanding the interaction between electromagnetic waves and thin crystals. If the following problems are well-addressed, the essential contribution of this paper is thought to be important for the development of light-active actuators.

1. The authors stated that the natural loss of the faceting of crystals during dissolution will result in the rounded shape of the cross-sections, but the manuscript has no image to show it. So the authors are suggested to give supplemented cross-section images.

2. The preparation method of a 2-μm thick crystal is too brief, the authors should supplement a detailed method in order to facile reproduction by others.

3. The propagation of light in Figure 6 and Figure 11 should be indicated clearly.

4. The scale bars should be added in Figure 1-3.

5. There are some typo errors on Page 3 “The introduction should briefly place …”. Please check them carefully.

Author Response

Response to the comments from Reviewer 2

Comment: 1. The authors stated that the natural loss of the faceting of crystals during dissolution will result in the rounded shape of the cross-sections, but the manuscript has no image to show it. So the authors are suggested to give supplemented cross-section images.

Response to the comment: Manipulations with the miniature crystal prepared by our method were very difficult and, unfortunately, the crystal was lost upon completion of the experiment and its removal from the experimental chamber. Therefore, we cannot provide images of this particular crystal. To make it easier to obtain images confirming the rounding of the edges of the crystal during thinning, we carried out the described procedure on a thicker crystal and took pictures showing that the dissolution proceeds more quickly on the edges of the crystal, leading to their rounding in the initial stages, and to obtaining a rounded section later on. dissolution. We have added an accompanying material containing the resulting images and following explanation in the text of the article:

“An example of a cross section image of a crystal that acquires a rounded shape after the etching is shown in Figure S1 of the Supplementary Materials.”

Comment: 2. The preparation method of a 2-μm thick crystal is too brief, the authors should supplement a detailed method in order to facile reproduction by others.

Response to the comment: Following text is added to the revised manuscript in the “Materials and Methods”

“The initial crystal of the complex, having an approximately square cross section ~10 µm thick, was glued to a glass capillary with a thin end (~40 µm in diameter). A cuvette with HCONH2 (formamide) used to thin the crystal was placed under optical microscope. Since [Co(NO3)5NO2]Cl(NO3) dissolves much more slowly in formamide than in water, formamide can work as a polishing etchant for these crystals. Another cuvette with dried dimethyl sulfoxide (DMSO) was placed nearby, which was used to quickly interrupt dissolution and wash the crystal from formamide. The crystal on the holder was carefully immersed in formamide, while rotational, vertical and horizontal movements were performed with constant monitoring of the dissolution process under the microscope. Dissolution occurs most rapidly from the edges of the crystal, and it acquires a rounded shape in cross section. Periodically, the dissolution process was stopped by immersion in DMSO to inspect and control the result. If necessary, the procedure is repeated until the desired thickness and geometry quality is achieved.”

Comment: 3. The propagation of light in Figure 6 and Figure 11 should be indicated clearly.

Response to the comment:  Arrows showing direction of the incident light is added to the figures and corresponding changes are made to their titles:

“The arrows show the direction of the incident light.”

Comment: 4. The scale bars should be added in Figure 1-3.

Response to the comment: Scale bars are added to the Figures 1-3 in the revised version.

Comment: 5. There are some typo errors on Page 3 “The introduction should briefly place …”. Please check them carefully.

Response to the comment: We apologize for accidentally leaving part of the text from the template in the manuscript. This part has been removed in the revised version.